



# Performance analysis of a Darrieus-type wind turbine for a series of 4-digit NACA airfoils

Krzysztof Rogowski[1], Martin Otto Laver Hansen[2], Galih Bangga[3]

[1]Institute of Aeronautics and Applied Mechanics, Warsaw University of Technology, Warsaw, 00-665, Poland
[2]Department of Wind Energy, Technical University of Denmark, Lyngby, DK2800, Denmark
[3]Institute of Aerodynamics and Gas Dynamics, University of Stuttgart, Stuttgart, 70569, Germany

*Correspondence to*: Krzysztof Rogowski (krzysztof.rogowski@pw.edu.pl)

**Abstract.** The purpose of this paper is to estimate the H-Darrieus wind turbine aerodynamic performance, aerodynamic blade loads and velocity profiles downstream behind the rotor. The wind turbine model is based on the rotor designed by McDonnell
Aircraft Company. The model proposed here consists of three fixed straight blades; in the future this model is planned to be develop with controlled blades. The study was conducted using the unsteady Reynolds averaged Navier-Stokes (URANS) approach with the k-ω shear stress transport (SST) turbulence model. The numerical two-dimensional model was verified using two other independent aerodynamic approaches: the vortex model developed in Technical University of Denmark (DTU) and the extended version of the CFD code FLOWer at the University of Stuttgart (USTUTT). All utilized numerical codes gave
similar result of the instantaneous aerodynamic blade loads. In addition, steady-state calculations for the applied airfoils were also made using the same numerical model as for the vertical-axis wind turbine (VAWT) to obtain lift and drag coefficients. The obtained values of lift and drag force coefficients, for a Reynolds number of 2.9 million, agree with the predictions of the experiment and XFoil over a wide range of angle of attack. The maximum rotor power coefficients are obtained at 0.5, which makes this impeller attractive from the point of view of further research. This work also addresses the issue of determining the
aerodynamic performance of the rotor with various 4-digit NACA airfoils. The effect of two airfoil parameters, maximum airfoil thickness and maximum camber, on aerodynamic rotor performance is investigated. Research has shown that if this rotor were to work with fixed blades it is recommended to use the NACA 1418 airfoil instead of the original NACA 0018.

## 1 Introduction

In 1931, G. J. M. Darrieus, a French aviation engineer, patented a wind turbine rotor capable of operating independently of the
wind direction and in adverse weather conditions. The Darrieus wind turbine having a rotor with a vertical rotation axis is often used to convert wind energy into electric energy. The Darrieus wind turbine is composed of several curved blades attached to a vertical rotating shaft. In 1927, G. Darrieus also suggested other possible solutions for turbines with a vertical rotation axis. One of them was H-rotor (rotor in H pattern), also known or "H-bar". The rotor of this type consists of long straight blades which are usually fastened to the tower by means of horizontal struts. Giromill is another type of wind turbine
with articulated and controlled straight blades to ensure maximization of energy extraction from the wind flow (Paraschivoiu,



2002), and the McDonnell 40 kW giromill is an example of such a construction (Fig. 1). Current work uses the geometry of the McDonnell Aircraft Company 40 kW design as a baseline for further research. The literature review made by the authors does not show any numerical research of this wind turbine. At the same time, Anderson et al. (1979) employing the Larsen Cyclogiro Performance Computer Program, estimated the device's maximum power factor of 0.51. The results of the power

curve obtained from the measurements of this rotor are also promising (Brulle, 1980). This is quite a high value even when compared to the performance of modern horizontal-axis wind turbines (HAWTs) (Hansen, 2015). The purpose of this work is to confirm the actual efficiency of this rotor, at first considering a rotor with fixed blades. Basic geometrical data of this rotor are given in Table 1.

| Parameter | Value |
|---|---|
| Rotor radius, R | 8.84 [m] |
| Blade length | 12.9 [m] |
| Chord, c | 0.61 [m] |
| Airfoil type | NACA 0018 |
| Number of blades, B | 3 |
| Rotor solidity, $\sigma = Bc/R$ | 0,2 |


**Table 1**. Basic geometric parameters of McDonnell 40 kW giromill

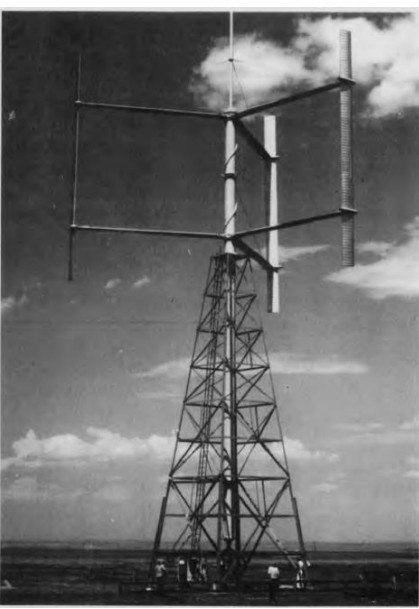

**Figure 1: Silhouette of the McDonnell 40 kW giromill, reproduced from Brulle (1980).**






## 1.1 VAWT employing NACA four-digit airfoil

4-digit NACA airfoils developed by National Advisory Committee for Aeronautics (NACA) describe the contour shape of an aircraft wing or a cross section of a wind turbine blade. The shape of the airfoil is described by means of a mathematical formula, given among others by Moran (2003). The shape of the 4-digit NACA airfoil series really depends on three

parameters: the maximum airfoil thickness, $t$, as a fraction of the chord (this value is expressed in the 4-digit airfoil denomination using the last two digits divided by 100); the maximum camber, $m$, ($100 \cdot m$ gives the first digit of the 4-digit airfoil denomination); the location of the maximum camber, $p$, ($10 \cdot p$ gives the second digit in the NACA XXXX airfoil denomination). For symmetrical airfoils, only the maximum thickness is needed for the complete airfoil description, the other two parameters, the maximum camber and the location of the maximum camber, are used to design asymmetrical profiles.

4-digit NACA symmetrical airfoils are probably the most commonly used foils in the design and manufacture of Φ-type and H-type Darrieus wind turbines. Vertical-Axis Wind Turbines (VAWT) Ltd company from Great Britain proposed a H-type rotor with straight blades also called a Musgrove rotor (Shires, 2013). The largest prototypes produced by this company were the VAWT-450 (rated power of 130 kW) and VAWT-850 (rated power of 500 kW); the first of them had the NACA 0015 airfoil and the second one had the NACA 0018 airfoil (Tjiu et al., 2015; Price, 2006; Mays et al., 1990). Other, more modern,

experimental studies of H-type rotors were conducted, among others, at Uppsala University. The 200 kW VAWT prototype with three straight blades was installed in early 2010 in Falkenberg, Sweden. This rotor, having NACA 0018 airfoils, provided 22.5 MWh of energy to the power grid during the tests within 1000 hours of operation (Apelfröjd at al., 2016; Ottermo et al., 2012). In the case of Φ-type rotors, in most commercial cases symmetrical airfoils of the 4-digit NACA series were used (Templin, 1979, Paraschivoiu, 2002; Tjiu et al., 2015; Möllerström et al., 2019).

Many authors have studied the aerodynamic performance of VAWTs equipped with one specific airfoil type, usually the NACA 00XX. Rezaeiha at al. (2019a) studied a 1-bladed vertical-axis wind turbine with the NACA 0015 airfoil using the scale adaptive simulation (SAS), URANS and hybrid RANS / LES methods. Their main purpose was to investigate the properties of dynamic flow detachment on the rotor blade at a low tip speed ratio. Another valuable numerical work of these authors was devoted to the research of three different small sizes wind turbines (Rezaeiha at al. 2019b). Zong at al. (2019)

proposed a method of passive control of dynamic stall of Darreius wind turbine blades. The authors proposed to install a small rod in front of the leading edge of the symmetrical NACA 0018 airfoil. Wong et al. (2018) performed a numerical analysis of a three-dimensional rotor having straight blades with NACA 0021 airfoils. They studied the effects of a flat plate deflector placed upwind of the vertical wind turbine. Elsakka et al. (2019) have found a method to calculate and store the local angle of attack for VAWT over the entire cycle from CFD. These authors examined the distribution of the local angle of attack of the

NACA 0015 airfoil blade. This could be also estimated by performing system identification, that is often used in aerodynamic modelling for time (Lichota and Agudelo 2016) or frequency domain (Lichota et al. 2019). Experimental and numerical testing of the three-bladed rotor was performed by Bianchini et al. (2017a). These authors confirmed the usefulness of the Shear Stress





Transport (SST) turbulence model to study the aerodynamic characteristics of the H-Darrieus rotor with blades having the NACA 0021 airfoils. Bianchini at al. (2015) and Bianchini at al. (2017b) studied the behavior of the symmetrical NACA 0018

airfoil in "Darrieus motion" and noticed that the results of the aerodynamic blade loads correspond to the cambered airfoil, this phenomenon was called "virtual camber effect" and comes from the fact that they operate in a curved inflow. Airfoil shape effect of a three-bladed rotor with a solidity of 0.1 was investigated by Mohamed et al. (2015) using many symmetrical and asymmetrical airfoil types. The symmetrical NACA 0015 airfoil showed equally high aerodynamic efficiency as the best among the tested airfoils. NACA 4-digit airfoils also provide good aerodynamic characteristics of Darrieus-type rotors working

in water (Hoerner et al., 2019). Sometimes a classic 4-digit NACA airfoil is the starting point in the airfoil shape optimization process (Ivanov et al., 2017).

The current work focuses on comparing the aerodynamic performance of a three-bladed H-Darrieus with 4-digit NACA airfoils depending on two parameters: the maximum airfoil thickness and the maximum camber. The shapes of the analyzed airfoils are shown in Fig. 2.


**Figure 2: Symmetrical 4-digit NACA series airfoils (a); cambered 4-digit NACA (b).**



## 1.2 Rotor aerodynamic performance

The operating rotor of the vertical-axis wind turbine creates aerodynamic loads that causes a local slowdown of the main flow.
The forces change with the azimuth, θ, that define the location of the rotor relative to the wind direction. In this work, azimuth zero means the position of the rotor in which the chord of the rotor blade is parallel to the main flow direction and the blade moves "against the wind". In present work, azimuth is measured from zero value as shown in Fig. 3. When the azimuth changes, the local angle of attack as well as local blade loads also change. The loads also depend on other factors, such as for example: the rotor solidity, the blade Reynolds number, airfoil shape and the tip speed ratio. The aerodynamic loads of blades are most
often represented as coefficients, defined as follows:

$$CF_{N,T} = \frac{F_{N,T}}{\frac{1}{2}\rho V_0^2 A},$$ (1)

where: $F_{N,T}$ are the aerodynamic blade load components, normal and tangential respectively; $V_0$ is the wind velocity; $\rho$ is the air density; A is the reference surface, in this work, A=c·1, where c is the length of the chord and 1 is the unit span of the blade. The tip speed ratio is defined as the ratio of tangential blade velocity $V_T$ and wind velocity:

$$TSR = \frac{V_T}{V_0} = \frac{\omega R}{V_0},$$ (2)

where: ω is the angular velocity of the rotor and R is the rotor radius.

In this work, the authors not only analyzed the distribution of aerodynamic blade loads but also the distributions of flow velocity in the aerodynamic wake behind the rotor. The velocity distribution is a direct result of the aerodynamic blade loads. In this work, two components of the velocity downstream behind the rotor were analyzed: a component parallel to the wind
direction, $V_x$, and a perpendicular component, $V_y$. These velocities were calculated using a rake that consists of 100 checkpoints and positioned at a distance of 1.5 R downstream behind the rotor axis of rotation, as shown in Fig. 4. The total length of the rake was 3 R, similarly as in the case of the experiments of Tescione et al. (2014) and Tescione et al. (2016). The velocity components $V_x$ and $V_y$ varies in time, but this paper presents their average values according to the formula:

$$V_{x,y} = \frac{1}{T}\int_0^T V_{x,y}(t)dt$$ (3)

where the averaging time, T, corresponds to the time of one full rotation of the rotor and the time integration step, is it equal to time step size.



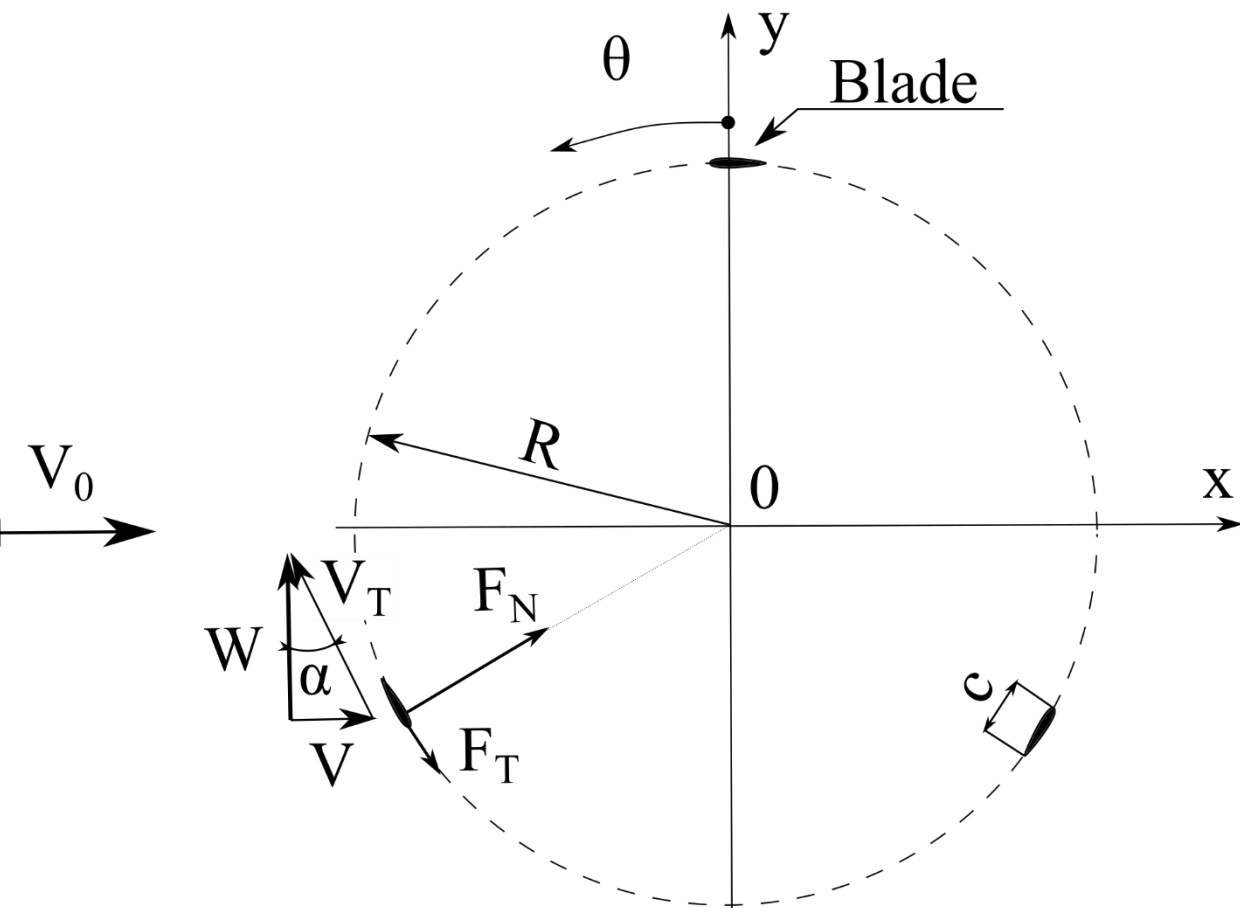

**Figure 3: Velocity vectors, aerodynamic blade loads and angles.**

## 2 Numerical model

All the numerical results of the Darrieus wind turbine aerodynamic characteristics presented in this article were obtained using the CFD tool - ANSYS Fluent. This section describes the geometric model, grid distribution and solver settings.

### 2.1 Geometric modelling

The calculations were carried out for the 40kW Giromill rotor presented in Section 1 and in Fig. 1. The real rotor has many elements that are not included in our numerical model. The current considerations did not take into account such structural elements as: blade supports as well as the rotating and fixed towers. The original rotor had controlled blades, but in this work only the rotor with fixed blades was investigated. In addition, the rotor was tested using a two-dimensional model consisting of three aerodynamic airfoils rotating around a fixed axis as depicted in Fig. 3. Rotor blades represented by two-dimensional NACA airfoils are treated as rigid bodies.





This rotor model was placed in a virtual wind tunnel (computational domain) that is large enough to eliminate the tunnel blockage effects, thus no tunnel correction model is needed. The rectangular computational domain with dimensions of 20 R by 60 R is presented in Fig. 4. Choosing computational domain sizes is one of the critical issues of modeling (Balduzzi et al., 2016), but based on following papers the domain size should be sufficiently large. Beri and Yao (2011) showed that a ratio of

domain width to rotor diameter of 6 is enough to correctly estimate the rotor power coefficient. Mohamad et al. (2011) studied the effect of computational domain size for a Savonius rotor performance and they showed that the ratio of the computational domain width to the Savonius rotor diameter of 10 gives sufficiently accurate rotor torque results. Rogowski (2018) received a similar conclusion when examining the influence of the square-length computational domain for a Darrieus rotor. Therefore, in our considerations presented in this paper, the ratio of domain width to rotor diameter was chosen to be 10. The distance

from the rotor axis to the inlet was assumed to be 5 D (Balduzzi et al., 2016). The distance between the rotor axis of rotation and the outlet is equal to 25 D, slightly larger than in the case of Ferreira et al. (2009) and Castelli et al. (2010) and smaller than in the case of Trivellato and Castelli (2014) or Castelli et al. (2011).

Transient flow around the rotating rotor was considered in this work. This issue requires the use of a special technique of moving meshes. The use of this technique, also known as the "sliding mesh" approach, requires the separation of a certain

domain area surrounding the rotating rotor. The distance between the common edge of both areas and the edge of the airfoils should be large enough due to possible numerical errors (Bangga et al., 2017). In our case, this area has the shape of a circle with a diameter of 2D as shown in Fig. 4.








**Figure 4: Numerical model, computational domain.**


**2.2 Numerical settings**

In these investigations, transient analyzes of the operating VAWT were performed, therefore, the Unsteady Reynolds Averaged Navier-Stokes (URANS) approach was utilized. ANSYS Fluent offers various techniques for solving transport equations depending on the problem being solved. Because the flow around the analyzed rotor is rather incompressible, the Mach number



is approximately 0.2, the pressure-based solver was utilized in this work. Previous works (Rogowski et al., 2018 or Rogowski and Hansen, 2019) have shown that the segregated algorithm available under the pressure-based solver provides a sufficiently accurate solution of aerodynamic characteristics of a VAWT, which is why it was also adopted in this work. This algorithm solves the governing transport equations sequentially, segregated from one another, moreover, the memory requirement for this algorithm is less than other algorithms. The pressure-based solver also offers the possibility of choosing different

velocity-pressure coupling methods and different discretization schemes. Following our own insights (Rogowski, 2018; Rogowski et al., 2018 or Rogowski and Hansen, 2019) and recommendations of ANSYS Documentation (ANSYS, Inc. Release 17.1), this work uses the standard SIMPLE pressure-velocity coupling algorithm, the second-order upwind spatial discretization schemes for all solving equations, whereas, gradients are computed according the least squares cell-based method.

As already mentioned, transient calculations were performed using the sliding mesh technique. This involves using an inner circular zone that rotates at the same angular velocity as the wind turbine rotor relative to the rectangular fixed outer zone. In this model, mesh nodes of the dynamic zone move rigidly. In addition, the rotating and stationary zones are connected with each other using a non-conformal interface. During the simulation of transient flow around an operating rotor the transport equations of momentum, continuity and turbulence are solved for defined time step. This time step size Δt is most often taken

constant and corresponds to a certain increase in azimuth Δθ=ω·Δt. The choice of time step size length is a particularly important issue in the numerical calculations of Darrieus rotor. According to the ANSYS Documentation (ANSYS, Inc. Release 17.1), the length of the time step size should be small enough for the required level of convergence of the solving equations to be obtained in the defined maximum number of iterations per time step. In addition, Rogowski (2019) showed that even if the assumed time step satisfies the desired convergence criterion, it may not be sufficient to capture all aerodynamic

phenomena, such as, for example, the effect of the tower's aerodynamic shadow on wind turbine blade loads. Rogowski (2019) and Rezaeiha et al. (2019a) agree that for a "clean rotor" (a rotor consisting only of blades) the length of the time step can be assumed equivalent to an azimuth increase, Δθ, of 0.1 deg. with the maximum iterations of 20 per time step, therefore, exactly such time step settings were adopted in these simulations.

Turbulence modeling is another key issue in the CFD approach. In the URANS approach two-equation turbulence models of

the k-ω and k-ε families are most commonly used, less often the one-equation Spalart-Allmaras turbulence model Balduzzi et al. (2016). This work is not dedicated to investigating the impact of the turbulence model on solving the problem of transient flow around a working rotor. Current simulations use a model widely used in this kind of flows (Rogowski, 2019), the Shear-Stress Transport (SST) k-ω model.

## 2.3 Computational mesh

The computational grid is a hybrid mesh composed of a structural mesh near the airfoil edges to resolve the viscous boundary layer and an unstructured mesh in the remaining area. Near the edges of the airfoils, rectangular elements were used, and in the case of unstructured mesh, triangular elements. All numerical models investigated in this work, have the same global grid



settings. The edges of the blades were divided into 200 elements of equal length. In the direction normal to the airfoil, the structural mesh has 60 layers with the growth rate of 1.12, and the thickness of the first layer near the airfoil edges of $5{,}3 \cdot 10^6$ m

which provides an average wall y+ of 0.50 for TSR=2 and of 0.53 for TSR=6. The growth rate of unstructured mesh elements is 1.04. The final grid shown in Fig. 5 consists of 234 840 elements and 136 059 nodes. The computational grid with global settings described above was used also by Rogowski (2019). This author shoved that these mesh settings provided sufficiently accurate results of the velocity fields and aerodynamic blade loads.

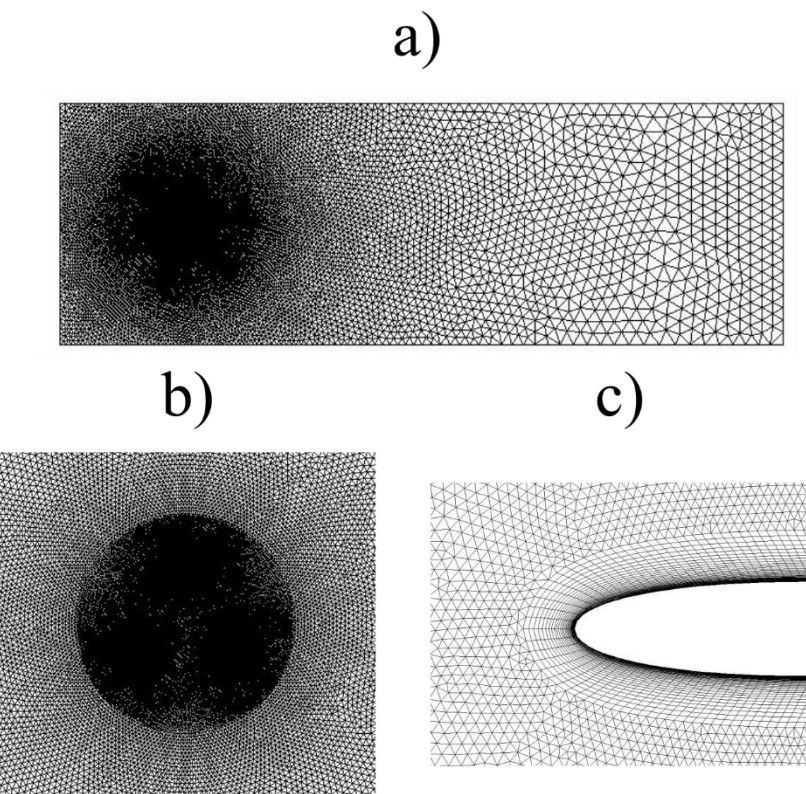


**Figure 5: Computational mesh, general view (a); rotor area (b); boundary layer mesh (c).**

**2.4 Mesh sensitivity study**

The discrete model described in section 2.3 has been tested to determine the number of elements that is required for accurate estimation of aerodynamic parameters. For this purpose, a series of additional simulations were carried out, in which the

number of equal airfoil edge divisions was changed. According to section 2.3, for all cases in this work the number of equal airfoil edge divisions was N = 200, but also calculations were carried out for three cases: N / 4, N / 2 and 2·N. A linear increase in the number of airfoil edge divisions also gives a linear increase in the total number of mesh elements from 121 350 for N / 4 to 425 892 for 2·N. However, the biggest difference in power coefficient, $C_P$, is for N / 4, i.e. for a number of divisions equal



to 50. An increase in the number of elements above 200 does not cause a significant change in $C_P$ but leads to an increase in
computational costs.

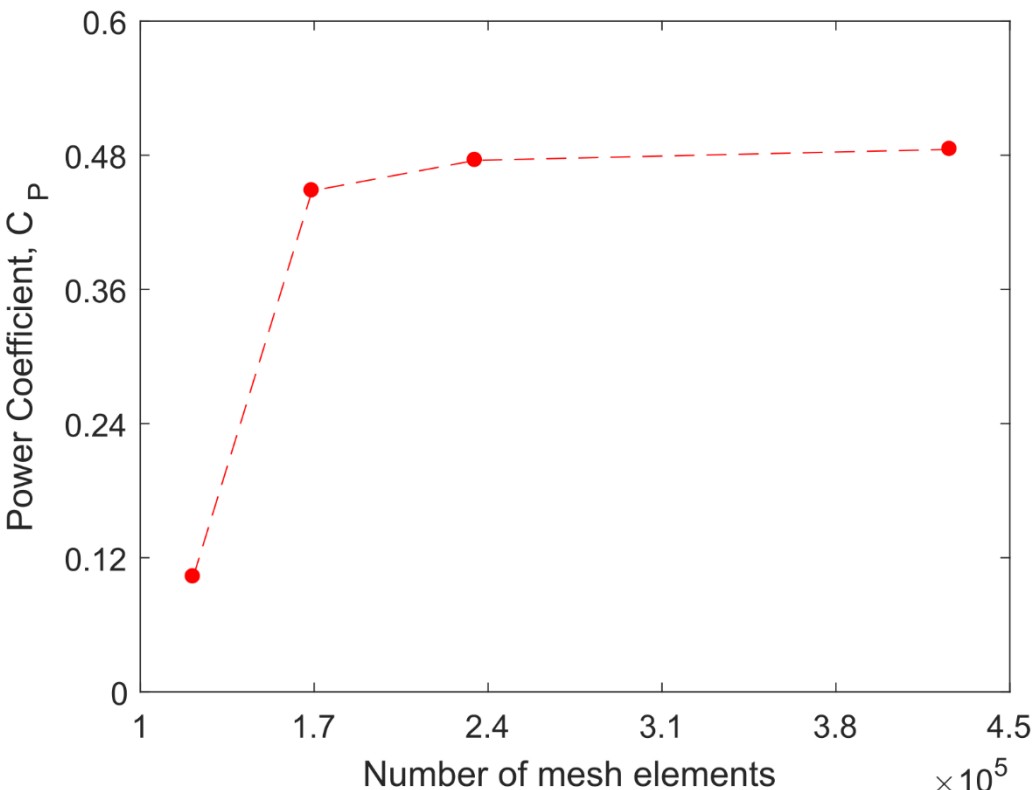

**Figure 6: Mesh sensitivity test, rotor power coefficient versus number of mesh elements.**

**2.5 Initial condition effects**

Before running numerical calculations, the initial conditions must be specified. Typically, homogeneous initial conditions are
assumed throughout the entire domain; in this work, the initial conditions corresponded to the flow parameters at the inlet.
During the simulation, transport equations are solved for each finite volume (see section 2.4) and the correct flow parameters
are determined. For each case studied, it is necessary to simulate several rotor revolutions to obtain a periodic result of the
aerodynamic blade loads and velocity distributions in each subsequent rotor rotation. This transient is the physical time it takes
the flow to become in equilibrium with the thrust force, i.e. the time to build up the induced wind speed.
Figure 7 shows the relation between the rotor torque coefficient, $c_m$, and time. The rotor torque coefficient is defined as:

$$c_m = \frac{Torque}{\frac{1}{2}\rho A_s R V_0^2},$$  (4)





where: $A_S$ is a rotor swept area, defined as $A_S=2R\cdot1$, where 1 is a unit span of the blade. The maximum values of the torque coefficient reach an almost constant values in each rotor revolution of about 1.3 after approximately 12 complete rotor revolutions corresponding to that an air particle moving with the free speed $V_o$ has travelled a distance of $\Delta x=12\cdot2\pi\cdot V_o/\omega=$ 220    $12\cdot2\pi\cdot R/TSR$, where TSR is the tip speed ratio. After 19 full turns, the differences are at the level of the second decimal place. The minimum values of this torque coefficient are about 0.3 after 12 full rotations of the rotor. A similar conclusion was made by Rogowski (2019), who studied the effect of initial conditions by simulating 50 full rotor revolutions. His simulation showed that 15 full revolutions of the rotor was sufficient to obtain repeatable results of both aerodynamic blade loads and velocity profiles behind the rotor.

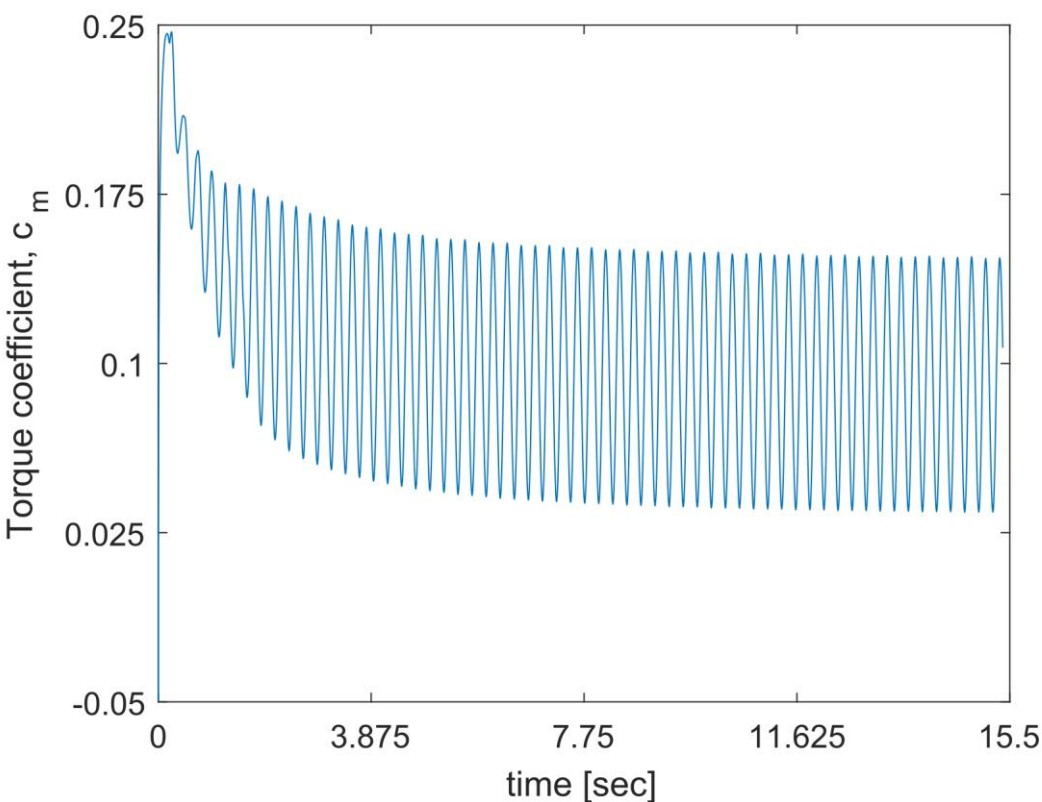


**Figure 7: Initial condition effect, rotor torque coefficient as a function of time.**

**2.6 Model Validation**

Validation of the numerical model described in Sections 2.1 and 2.2 was performed in two ways. First, the lift and drag airfoil characteristics from the experiment of Sheldahl and Klimas (1981) were taken into account as a baseline for steady-state CFD 230    simulations. Then, the instantaneous normal and tangential aerodynamic blade loads computed using ANSYS Fluent CFD code were compared with the results obtained by the vortex model and FLOWer CFD code.





### 2.6.1 Lift and drag airfoil characteristics

Numerical calculations of aerodynamic force coefficients, lift and drag, for one rotor blade airfoil were carried out using the same computational mesh (Fig. 5) as in the case of transient calculations of the entire rotor; the same turbulence model was

also used as in the case of transient analysis. During steady-state numerical simulations, both parts of the computational mesh shown in Fig. 5 remained stationary with respect to each other; the angle of attack of the analyzed airfoil remained unchanged during the simulation. A numerical experiment was carried out for the blade Reynolds number, defined as (Paraschivoiu, 2002):

$$Re = \frac{R\omega c}{v_\infty},\tag{5}$$

where: R is a rotor radius, ω is an angular velocity of the rotor; c is a chord length and $v_\infty$ is kinematic viscosity of the air. This Reynolds number was used to determine the undisturbed flow velocity $V_0$ for steady-state simulations. In these investigations, it was assumed that the angular velocity of the rotor was constant for all tip speed ratios and equal to 8.18 rad / s. Knowing that the rotor radius is 8.48 m, the tangential velocity of the rotor blade is 69.4 m / s for the Reynolds number to be equal to 2.9 million. Aerodynamic force coefficients, lift coefficient $C_L$ and drag coefficient $C_D$ are defined as:

$$C_{L,D} = \frac{L,D}{\frac{1}{2}\rho V_0^2 A},\tag{6}$$

where: L and D are lift and drag forces, ρ is air density, A is reference area (in this investigations A=c·1, where 1 is unit span of an infinitely long rotor blade). Numerical results of these coefficients of aerodynamic forces were compared with experimental studies of Sheldahl and Klimas (1981) and with the results obtained from the XFoil program; these results are shown in Fig. 8.

Discussing the presented results, it can be stated that both lift force coefficients as well as drag coefficients obtained from CFD are consistent with the experiment in the range of angles of attack from 0 to 14 degrees in the case of lift force and from 0 to 16 degrees in the case of drag force. From the zero lift, the $C_L$ curve is practically a straight line up to the angle of attack of 12-13 deg. The aerodynamic derivative ($dC_L/d\alpha$) of the experimental $C_L$ curve for the angle range of 0-12 degrees is 5.681 per rad. The aerodynamic derivative of the $C_L$ curve from CFD is larger by 3.4% compared to the experiment, whereas the

aerodynamic derivative from XFoil is larger by 11% compared to the experiment. The minimum drag coefficient corresponds to the zero angle of attack and is 0.0076 for the experiment, 0.0061 for XFoil and 0.01 for CFD. The somewhat larger minimum drag for the CFD is most likely because transition was not considered.





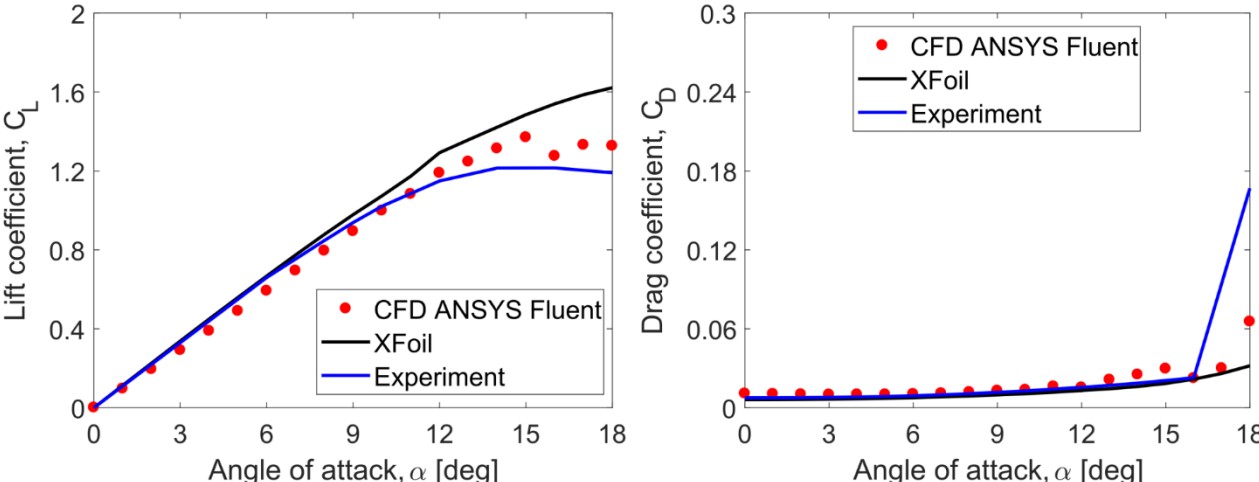

**Figure 8: Aerodynamic airfoil characteristics: lift coefficient (on the left), drag coefficient (on the right). The comparison of CFD results with the experiment by Sheldahl and Klimas (1981) and with the XFoil predictions.**

### 2.6.2 Instantaneous aerodynamic blade loads

All numerical results presented in this paper were computed using ANSYS Fluent solver and the unsteady RANS (URANS) method with the SST k-ω turbulence model. For the comparison, the aerodynamic blade loads were also computed using a

CFD code FLOWer from German Aerospace Center that was continuously extended for wind energy applications at IAG (Weihing et al., 2019) and a vortex model from Technical University of Denmark (DTU). The vortex method calculates the induction at the blade positions from the influence of the continoulsy developing wake vorticity using the Biot-Savart law. The strength of the instantaneous bound circulation was calculated from the local inflow and airfoil data using the Kutta-Joukowski theory and a new vortex was released in every time step at the trailing edge to ensure that the total circulation remains constant

in time. The airfoil data were calculated using XFoil and presented in Fig. 8. However, the effect of decambering from the blades moving in a curved was estimated as approximately 1 degree using thin airfoil theory and effectively shifting the lift curve 1 degree to the left. No dynamic stall model was used here. In these investigations, the rotor operating at the tip speed ratio of 5 and equipped with NACA 0018 airfoils was analysed. The normal and tangential blade loads are shown in Fig. 9. The two employed CFD codes and the vortex model deliver similar results of aerodynamic blade loads and taken as verification

that the equations are solved correctly.





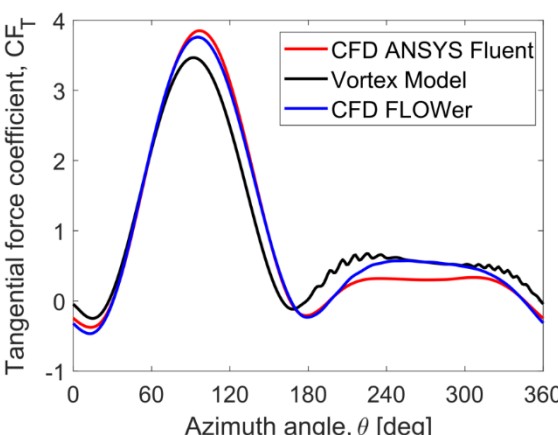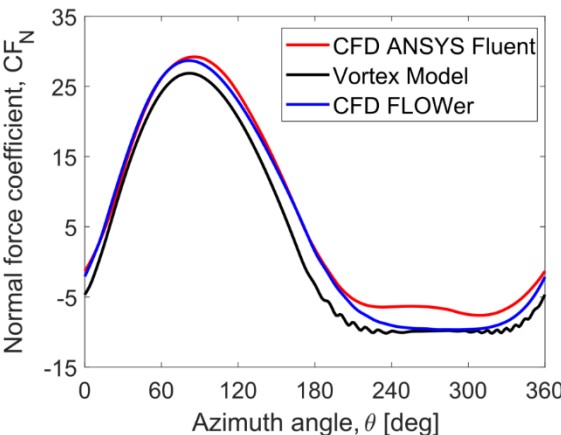

**Figure 9: Aerodynamic blade load components, tangential component (on the left) and normal component (on the right). Comparison of numerical results obtained with three different independent aerodynamic codes.**

## 3 Results and Discussion

This section summarizes the results of studies on the aerodynamic performance of a Darrieus-type wind turbine. The first two subsections discuss the aerodynamic blade loads for symmetrical and cambered NACA airfoils. The last subsection describes aerodynamic wake downstream behind the rotor.

### 3.1 Aerodynamic blade loads for symmetrical 4-digit NACA airfoils

Aerodynamic blade loads, tangential and normal components, depend on various aspects such as, for example, tip speed ratio, rotor geometry and Reynolds number. This paper only takes into account the influence of the airfoil shape and the tip speed ratio on aerodynamic blade loads. Figure 10 shows the distributions of the non-dimensionalized aerodynamic blade load components for two tip speed ratios: 2 and 5. The results are given for four symmetrical NACA airfoils from NACA 0012 to NACA 0021. This figure clearly shows that in the case of symmetrical airfoils, the tip speed ratio has the greatest impact on the tangential blade load component. In the upwind part of the rotor, thinner airfoils show faster loss of tangential force with azimuth. In the case of the NACA 0012 airfoil, this decrease is already at $\theta$=52 degrees causing the maximum value of this force coefficient to drop by almost 66% compared to the NACA 0018 airfoil. The early decrease in the tangential force in the case of thinner airfoils is related to the airfoil characteristics in vicinity of the critical angle of attack (the angle of attack that produces the maximum lift coefficient). Analyzing steady-state experimental data of lift coefficients of NACA 0012, NACA 0015 and NACA 0018 obtained by Sheldahl and Klimas (1981) for Reynolds number of $1 \cdot 10^6$ it can be stated that the airfoil with a relative thickness of 12% achieves the highest value of the maximum lifting force coefficient, however, the critical angle of attack is smaller compared to the NACA 0018 airfoil. In addition, in the case of the NACA 0012 airfoil, the decrease in lift above the critical angle is much more rapid, indicating a leading edge stall, than in the case of NACA 0015 and





NACA 0018 airfoils, for which the decrease in lift is much milder and where the separation starts at the trailing edge and
gradually increasing with the angle of attack. It can also be seen from Fig. 10 that the tangential blade load coefficient decreases
rapidly in the downwind part of the rotor at azimuth around 200-235 degrees deepening on the relative thickness of the airfoil.
In the case of low tip speed ratios, local angle of attack on both rotor sides exceeds critical values. Therefore, local decreases
in tangential force observed in this part of the rotor result from exceeding the critical angle of attack. The differences between
aerodynamic blade load coefficients for higher tip speed ratio of 5 are practically invisible to the naked eye. In general, all
analyzed airfoils give an almost identical distribution of aerodynamic blade loads. However a subtle difference can be seen for
an azimuth of about 95 degrees where the peak of the tangential blade load first increases with the thickness of the airfoil and
then decreases. Song et al. (2019), who also analyzed the aerodynamic performance of a Darrieus rotor with higher solidity
and with symmetrical NACA 00XX airfoils, obtained a similar correlation. In the optimum tip speed ratio range, the maximum
power coefficient was achieved by the rotor with NACA 0015 airfoils.
From the developer's point of view, the average tangential aerodynamic blade load component is much more interesting. For
this purpose, a filled contour plot containing the isolines of the rotor power coefficient as a function of tip speed ratio and
maximum airfoil thickness was created (Fig. 11). This figure clearly shows that the optimal airfoil for the examined wind
turbine is the NACA 0018 airfoil. It should be emphasized, however, that the area of the largest power coefficients (the area
of $C_P$ around 0.5) for the NACA 0015 airfoil shifts towards higher tip speed ratios. Moreover, the thicker airfoil (NACA 0018)
is slightly worse at a tip speed ratio of 6 than the NACA 0015 airfoil.



**Figure 10: Aerodynamic blade load components, tangential component (on the left) and normal component (on the right), for different maximum thickness of the NACA 4-digit airfoil and for two tip speed ratios 2 and 5.**



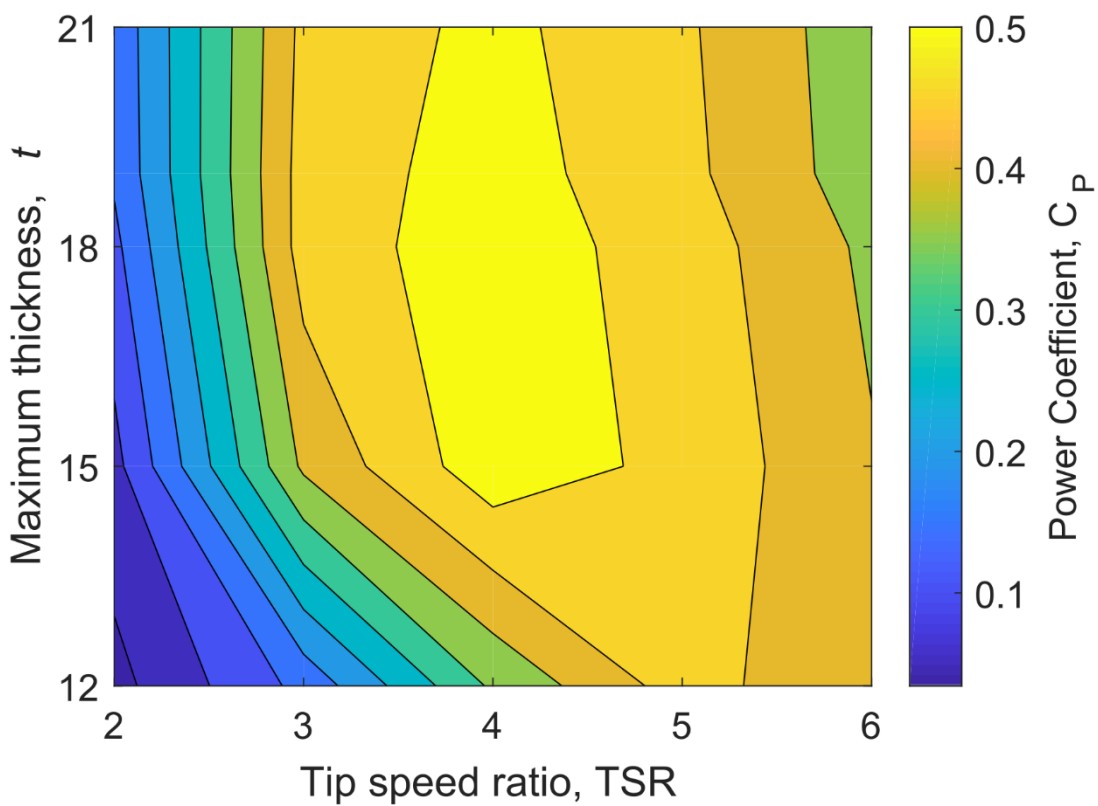

**Figure 11: Rotor power coefficient as a function of maximum airfoil thickness and tip speed ratio.**

### 3.2 Aerodynamic blade loads for cambered 4-digit NACA airfoils

The previous section discusses the impact of maximum airfoil thickness on rotor performance. Another very important parameter of NACA 4-digit airfoils is the maximum airfoil camber. The impact of this factor on rotor performance is discussed in this section.

Figure 12 illustrates the impact of the maximum camber on aerodynamic blade loads for two tip speed ratios, 2 and 5. Generally, as the power coefficient results presented in Fig. 13 show, the airfoil with a small camber equal to 1% (NACA 1418) shows better performance over the entire analyzed tip speed ratio spectrum than the symmetrical airfoil NACA 0018. The shape of aerodynamic blade load components of curved airfoils is also different compared to symmetrical airfoils (Fig. 12). This is most visible for tangential blade load component at a tip speed ratio of 2. Asymmetrical airfoils do not experience a sudden loss of tangential blade load for azimuths 70 deg and 230 deg. The increase in the maximum camber of the airfoil causes a decrease in the maximum tangential force on the upwind part of the rotor and an increase in the maximum tangential force on the downwind side. For the upwind part of the rotor, the maximum tangential blade load coefficient for the NACA 4418 airfoil is 15% lower compared to the NACA 1418 airfoil and 11% compared to the NACA 0018 airfoil. For the downwind





part of the rotor, the maximum tangential blade load for the NACA 1418 airfoil is 21% lower compared to the NACA 4418 airfoil. The shape of the tangential blade load curves changes when the tangential blade velocity increases compared to the wind speed. Figure 12 shows the relationship of the tangential blade load component of the blade for a ratio of these velocities of 5. The results are given for five 4-digit NACA series airfoils. The differences in the maximum values of the tangential blade load coefficient for different NACA airfoils on both sides of the rotor are much larger than for the velocity of 2. For the upwind

part of the rotor, the maximum tangential blade load coefficient value for the NACA 4418 airfoil is 1.94 and is 49.5% lower compared to the NACA 0018 airfoil. In the case of the downwind part of the rotor, the maximum tangential load coefficient for the NACA0018 airfoil is 0.32 and is 78% lower compared to the NACA 4418 airfoil. The change in the maximum values of the tangential blade load coefficient as a function of the maximum airfoil camber as percentage of the chord for the upwind part of the rotor is almost linear with the slope of -0.47. Interestingly, all $CF_T$ curves have the same value close to zero for an

azimuth close to 180 degrees. This is the location where the chords of the airfoils are parallel to the wind direction and the wind blows from the back. All curves also have a very similar value of about -0.38 when the blade passes through an azimuth of 13 degrees. The ratio of the maximum values of the tangential blade load for the upwind part of the rotor to the maximum values of this load for the downwind part as a function of maximum airfoil camber can also be approximated by a straight line. This ratio reaches a value from 11.6 for the NACA 0018 airfoil to 1.34 for the NACA 4418 airfoil. Thus, for the airfoil with

the largest maximum camber examined, the tangential blade load peak on both sides of the rotor area is almost equal.

While comparing the distribution of the normal component of the aerodynamic blade load, it can be seen that the differences in these distributions increase with tip speed ratio. For a tip speed ratio of 2, $CF_N$ distributions for the different airfoils investigated are very similar, whereas, for a tip speed ratio of 5 the curves are very similar, but they are offset by almost constant value from one another. A detailed analysis of the normal component of the aerodynamic blade load for a tip speed

ratio of 2 shows, however, that, as in the case of TSR = 5, the behavior of the curves is identical, the curve offset is also visible, although the value of the offset of the curves is very small. Only the curve corresponding to the NACA 0012 airfoil has a different shape from the others and as already described, it is associated with a sudden loss of lift on this profile. It can be seen that in the 0-60deg, 164-226deg and 314-360deg azimuth ranges, the trends of both CFT and CFN for TSR = 2 are identical to those for TSR = 5. In the upwind part of the rotor, the absolute value of the maximum normal blade load coefficient decreases

almost linearly with the maximum camber of the airfoil from 29.2 for the NACA 0018 airfoil to 17.7 for the NACA 4418 airfoil, whereas, the $CF_{Nmax}$ in the downwind part of the rotor increases almost linearly from 7.63 for the NACA 0018 airfoil to 20.13 for the NACA 4418 airfoil. The slopes of straight lines passing through $CF_{Nmax}$ as a function of maximum airfoil camber are -2.865 for the upwind part of the rotor and 2.993 for the downwind part of the rotor. In the case of TSR=2, the slops are also very similar, -0.591 for the upwind part of the rotor and 063 for the downwind part of the rotor, however, in the

case of TSR = 2, the NACA 0018 profile was not used for the analysis. For TSR = 5 the ratio of maximum normal blade load coefficient in the upwind part of the rotor to the downwind part of the rotor is from 3.82 for the NACA 0018 airfoil to 0.88 for the NACA 4418 airfoil, while for TSR = 2 this ratio is from 1.72 for the NACA1418 airfoil to 1.2 for the NACA 4418 airfoil.





**Figure 12: Aerodynamic blade load components, tangential component (on the left) and normal component (on the right), for different maximum camber of the NACA 4-digit airfoil and for two tip speed ratios 2 and 5.**





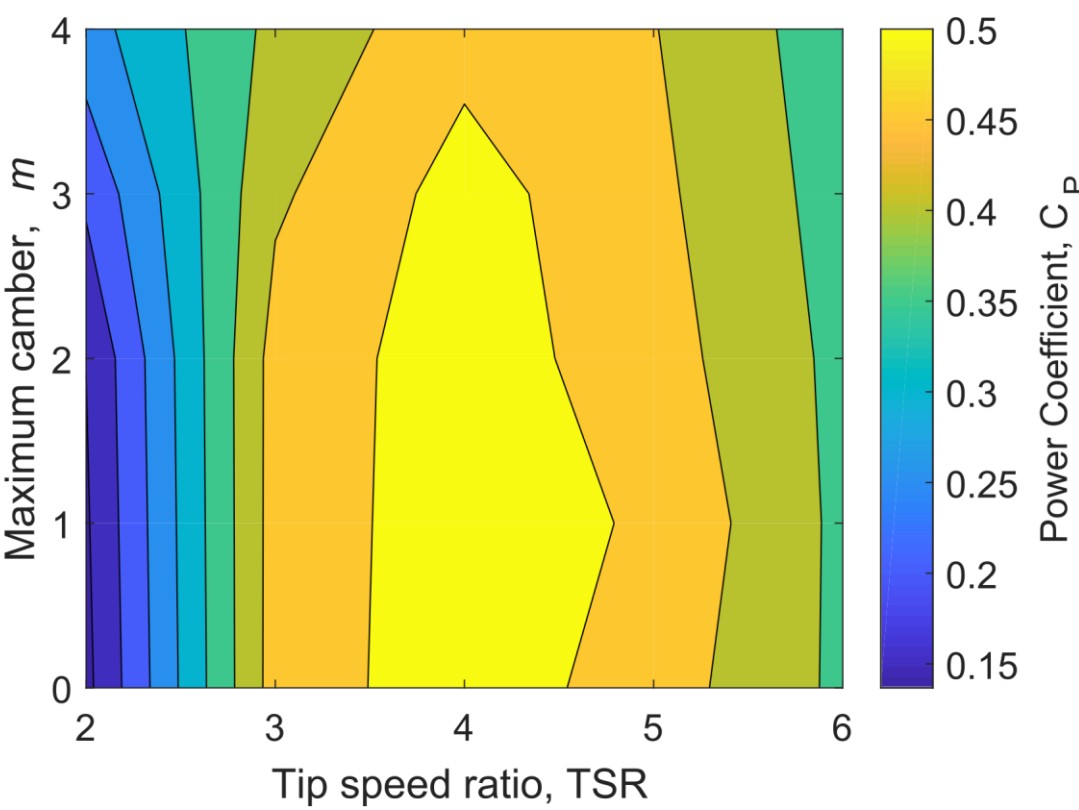

**Figure 13: Rotor power coefficient as a function of maximum airfoil camber and tip speed ratio for t/c=10%.**

### 3.3 Aerodynamic wake for symmetrical and cambered 4-digit NACA airfoils

The different shape of the aerodynamic blade loads described in the previous two subsections also affects the velocity distribution in the aerodynamic wake behind the rotor. This section deals with the impact of symmetrical airfoils on the air velocity field 1.5 R downstream, measured from rotor axis, of the rotor (Fig. 4).

Figure 14 shows the impact of tip speed ratio on the distribution of two velocity components, $V_x$ and $V_y$, downstream behind the rotor. Both velocity components are related to the wind velocity $V_0$ while the coordinate y (please see Fig. 4) is normalized

by the radius of the rotor R. The velocity results presented in Fig. 14 are given for the NACA 0018 airfoil. It is easy to see that the rotor has a greater effect on the velocity component $V_x$ than on the component perpendicular to the wind direction $V_y$. As tip speed ratio increases, the velocity $V_x$ decrease is larger. The average velocity of each of these profiles decreases almost linearly as a function of tip speed ratio from 0.92 for TSR = 2 to 0.57 for TSR = 6. It also seems that except in the case of TSR = 2, the velocity component $V_x$ profiles are almost symmetrical. In the case of velocity component $V_y$, its average for each tip

speed ratio value is very close to zero. When analyzing the distribution of this velocity component, the more important factor visible to the naked eye is the slope of the function $V_y / V_0$ (y / R). In this discussion, we have limited our considerations for

the y / R ratio range from -1 to 1 because in this range the $V_y/V_0$ curves are the most linear. For tip speed ratio 2 the curve

slope $d(V_y/V_0)/d(y/R)$ is small and is 0.014, whereas, for tip speed ratio 5 the slope is 0.0515. It is easy to see from Fig. 14

that the change in the slope of these curves with tip speed ratio is not linear. In order to better illustrate this, a graph of the

slope as a function of tip speed ratio was prepared - Figure 15. From this figure, it can be seen that as the tip speed ratio

increases, the increase in slope is getting smaller.

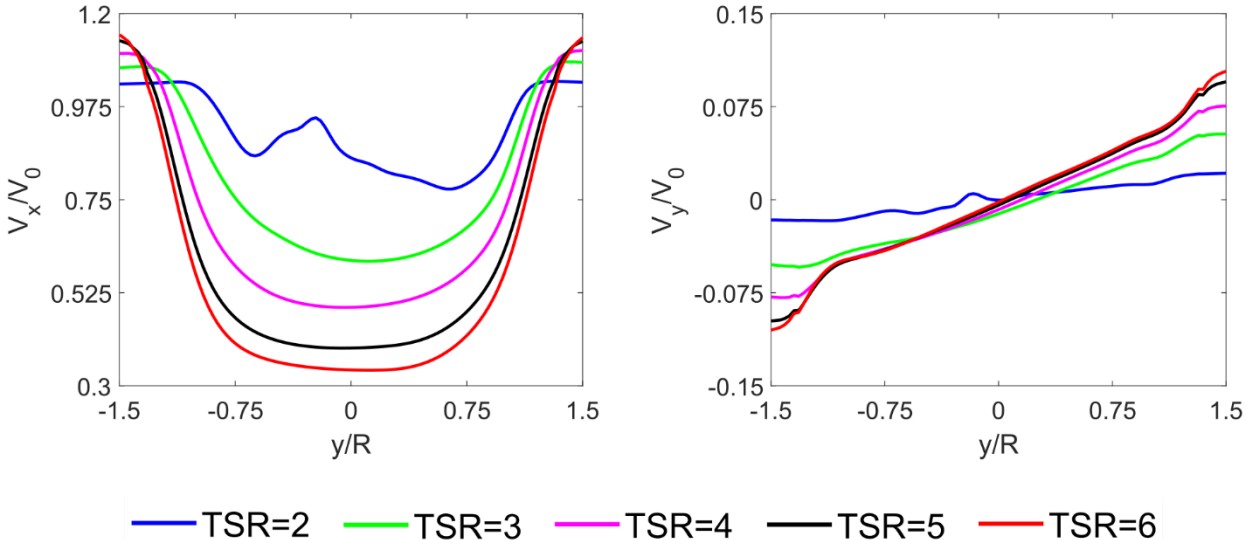

**Figure 14: Velocity profiles downstream behind the rotor at the distance of 1.5 R form the rotor axis of rotation; velocity component parallel to the wind direction, $V_x$, (on the left) and perpendicular to the wind direction (on the right). The results are given for the**
**NACA 0018 airfoil and for five tip speed ratios. Velocities $V_x$ and $V_y$ are normalized by wind speed $V_0$.**

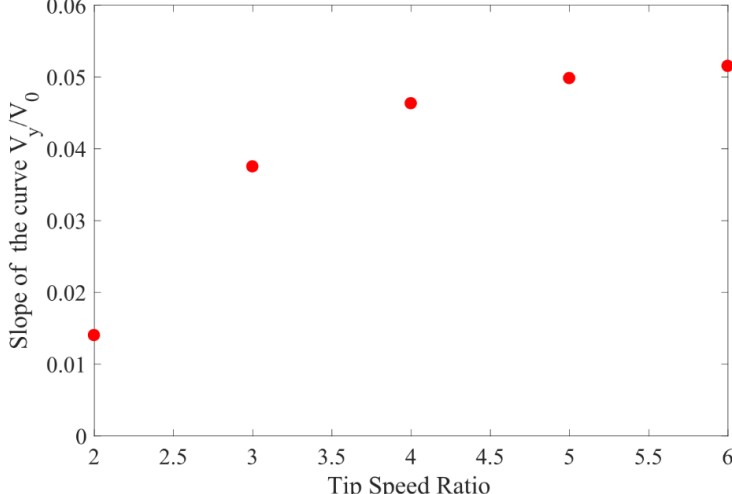

**Figure 15: The slope of the $V_y/V_0$ function relative to y/R depending on the tip speed ratio.**




Generally, as can be seen from Fig. 14, the value of velocity component $V_y$ is very small compared to the component $V_x$. The average value of velocity component $V_y / V_0$ of all velocity profiles is 0.0028 and is smaller by about 99.6% compared to the
average $V_x / V_0$ of all velocity profiles. To better show the share of both components of velocity ratios, a vector graph of velocity ratio $V/V_0$ was made for two tip speed ratios values 2 and 6 (Fig. 16). The speed V is defined as the root of the sum of the squares of the components $V_x$ and $V_y$. The figure clearly shows that the velocity vectors for TSR = 6 are more deflected outwards at the coordinate $| y / R | = 1.5$ than in the case of TSR = 2.

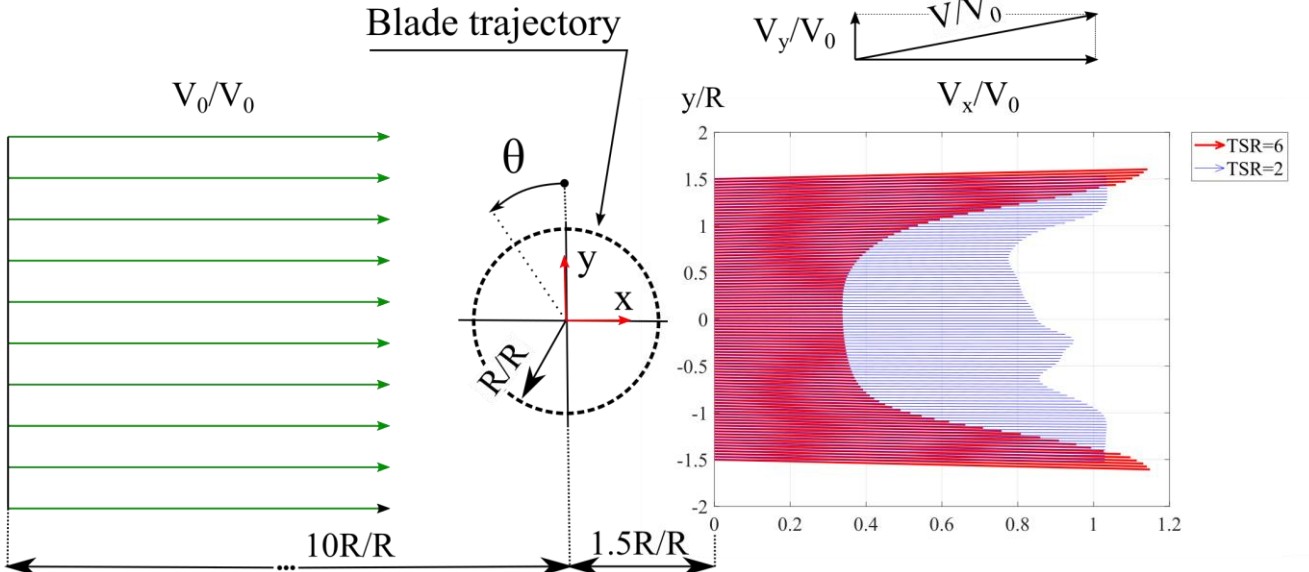

**Figure 16: The effect of tip speed ratio on the $V_y$ velocity component.**

Above, the influence of tip speed ratio on the distribution of velocity downstream behind the rotor has been described for NACA 0018 airfoil. Figure 17 shows the distribution of the velocity components $V_x$ and $V_y$ in the aerodynamic wake downstream behind the rotor for two tip speed ratios 3 and 6 depending on the relative thickness of the airfoil. Velocity distributions are given for four symmetrical airfoils: NACA 0012, NACA 0015, NACA 0018 and NACA 0021. This figure
clearly shows that, except for the NACA0012 airfoil at the tip speed ratio of 3, the differences in the velocity profiles downstream behind the rotor, both the $V_x$ and $V_y$ components, are negligible. The average value of all velocity profiles $V_x/V_0$ for TSR = 3 is 0.81 with a standard deviation of 0.014, whereas, in the case of the velocity profiles $V_y/V_0$ the average value is -0.0049 with a standard deviation of 0.0023. For a tip speed ratio of 6 the average velocity of all $V_x / V_0$ velocity profiles is 0.01173 with a standard deviation of 0.0117. The average value of the second velocity component for all $V_y / V_0$ velocity
profiles is 0.0004 with a standard deviation of 0.001. Based on the average velocity values and the standard deviation, it can be stated that the share of the velocity component parallel to the wind direction is larger than in the case of the velocity component $V_y$ and that the differences in the velocity profiles downstream behind the rotor are lower in the case of the velocity component $V_y$.




**Figure 17: Velocity profiles downstream behind the rotor at the distance of 1.5 R from the rotor axis; velocity component parallel to the wind direction, $V_x$, (on the left) and perpendicular to the wind direction (on the right). The results are shown for different maximum thickness of the NACA 4-digit airfoil and for two tip speed ratios 3 and 6.**

Figure 18 presents the shapes of the velocity profiles downstream of the rotor depending on the maximum airfoil camber and

tip speed ratio. The results are shown for three airfoils: NACA 0018, NACA 2418 and NACA4418. Analyzing the velocity

distributions shown in Fig. 18, some similarities can be noticed between the curves for TSR = 3 and TSR = 6. The $V_x / V_0$

curves seem to be very close to each other while the $V_y / V_0$ curves seem to be offset by some fixed value. Analyzing the

average values and standard deviations, it can be seen that both in the case of the velocity component $V_x$ and $V_y$ the differences

between the results are very small, both for the case TSR = 3 and TSR = 6. For TSR = 3, the average velocity of all three

$V_x / V_0$ velocity profiles is 0.57 and the standard deviation is 0.007; the average velocity for all $V_y$ velocity profiles is -007

with a standard deviation of 0.0068. In the case of TSR = 6, the average velocity of the three average $V_y / V_0$ velocity profiles

is 0.797 with the standard deviation equal to 0.0054, whereas, in the case of the second velocity component, the average





velocity is -0.012 with standard deviation of 0.0052. The slopes of the $V_y / V_0$ curves in the range of $y / R$ from -1 to 1 are almost identical and are on average 0.037 for TSR = 3 and 0.05 for TSR = 6.


Figure 18: Velocity profiles downstream of the rotor at the distance of 1.5 R form the rotor axis; velocity component parallel to the wind direction, $V_x$, (on the left) and perpendicular to the wind direction (on the right). The results are shown for different maximum airfoil camber of the NACA 4-digit airfoil and for two tip speed ratios 3 and 6.

**4 Conclusion**

The purpose of these investigations was to analyze the impact of two parameters of 4-digit NACA series airfoils, maximum airfoil thickness and maximum camber on the aerodynamic blade load, on aerodynamic efficiency (maximum rotor power coefficient) of the H-Darrieus rotor and velocity distribution downstream behind the rotor. Both aerodynamic blade loads and velocity profiles downstream behind the rotor depend on these two geometrical airfoil parameters as well a tip speed ratio.

Different numerical methods of fluid dynamics were used in this work and for CFD the k-ω SST turbulence model was utilized.





- Steady-state simulations confirmed that the numerical model and computational mesh give reasonable results of aerodynamic force coefficients, lift and drag components. ANSYS Fluent better predicts the relationship between lift and angle of attack, while XFoil gives a slightly better result of a minimum drag coefficient compared to the experiment. Other numerical codes like FLOWer CFD and vortex model have shown that ANSYS Fluent CFD code correctly estimates the unsteady blade load components of the wind turbine rotor.

- The first transient investigations concerned symmetrical airfoils from NACA 0012 to NACA 0021. When considering a wind turbine equipped with symmetrical NACA 0018 airfoils, the best aerodynamic performance was observed in the majority of tip speed ratio ranges.

- Although the NACA 0012 airfoil has the largest maximum lift coefficient of all symmetrical airfoils tested, it gives the worst results of the tangential blade load in the low tip speed ratio range. This is due to the worse airfoil characteristics in the detachment area compared to thicker airfoils.

- The analysis showed that symmetrical airfoils are much worse at low tip speed ratios. This is because of the worse characteristics of these airfoils in the stall regime. The introduction of one percent maximum camber greatly improves the aerodynamic performance of the rotor over the entire tip speed ratio range.

- The effect of the relative airfoil thickness on the characteristics of aerodynamic blade load components is larger at low tip speed ratios, whereas, the maximum camber affects more these characteristics at higher tip speed ratios.

- The use of cambered airfoils should improve the dynamic properties of the structure, e.g. reduce vibration. In the case of the NACA 4418 airfoil, the ratio of the maximum tangential blade load for the upwind part of the rotor to the downwind part is 88% lower compared to the NACA 0018 airfoil.

- The study examined the impact of tip speed ratio on the velocity distribution in the aerodynamic wake of a rotor equipped with NACA0018 airfoils. Numerical analysis showed that as the tip speed ratio increases, there is a linear decrease in the average velocity $V_x$ (velocity component parallel to the wind direction) of these profiles. In the case of the transverse velocity component $V_y$, its average for each tip speed ratio value is very close to zero. In the case of a rotor equipped with the symmetrical profile NACA 0018, it was observed that the share of the velocity component $V_y$ in the aerodynamic shadow of the rotor is very low in the entire tested tip speed ratio range.

- The increase in the relative thickness of symmetrical airfoils does not cause significant differences in the velocity distribution downstream behind the rotor in the entire investigated tip speed ratio range. The impact of maximum airfoil camber on the velocity distribution in aerodynamic shadow of the rotor is negligible. As in the case of symmetrical airfoils, the tip speed ratio has the biggest influence on the velocity distribution in the aerodynamic wake downstream behind the rotor.



Data availability.

A commercial ANSYS Fluent tool was used in the work. The use of this tool requires the purchase of an appropriate license. Autor KR will be happy to share all generated data for comparison purposes.

Author contributions.

KR - main author of the text; performing all calculations with ANSYS Fluent

MOLH - editorial and substantive correction of the article; transient calculations with a vortex model; calculation of profile characteristics using the XFoil program

GB - editorial and substantive correction of the article; transient calculations using FLOWer CFD code

Competing interests.

The author declares that he has no conflict of interest.

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
