# Peer review of "Performance analysis of a Darrieus-type wind turbine for a series of 4-digit NACA airfoils"

_Wind Energy Science, 2019_

## Referee Comment (RC1) · Anonymous Referee #1 · 26 Jan 2020

Interesting work. Focused on the description of the impact of changes in thickness and camber of the H-turbine blade profile on its aerodynamic properties. Changes to these characteristics are discussed without analyzing changes in the flow structure. Perhaps a separate work is being devoted by the Authors to the analysis of changes in flow structures.

General thoughts. The paper presents a rational test of the influence of the mesh size based on the number of nodes on the edge of the profile, and not on the total number of nodes.

The authors ignore the influence of the Reynolds number, but at constant wind speed, two TSR values mean two Re values at the same time. In the analyzed range of TSR values, this means a three-fold increase in Re between TSR = 2 and 6.

To explain the reasons for the change in turbine performance characteristics, it may be necessary to pay attention to the following effects. For TSR = 2, the angle of attack change is + - 30 degrees, which means fow separation. For TSR = 5, the max angle of attack change is + - 11.6 degrees, which means work without flow separation. For TSR = 2, with the blade rotation angle of 46 degrees, the profile approach angle of attack 15 degrees corresponding to the flow separation. For TSR = 3, at a blade rotation angle of 66 degrees, the angle of attack of the profile exceeds 15 degrees corresponding to the flow separation. For TSR = 4 and larger TSRs, the profile angle of attack never exceeds the critical angle of attack. The influence of flow inertia effects is not strong. Increasing TSR means only slightly increasing the rate of the angle of attack changes.

It is seen lack of basic characteristics for different profile thicknesses, although the authors provide some information. "In addition, in the case of the NACA 0012 airfoil, the decrease in lift above the critical angle is much more rapid, indicating a leading edge stall, than in the case of NACA 0015 and NACA 0018 airfoils, for which the decrease in lift is much milder and where the separation starts at the trailing edge and 300 gradually increasing with the angle of attack. "

The description is focused on the analysis of force variation. And yet the course of the curves is the result of changes in the flow structure. The changes in the flow structure cause certain effects. The symptoms are described, not the causes.

There are a number of questions that are not answered.

The dramatic change in the performance characteristics for TSR = 2 is surprising. When the camber changes, the transfer of received energy from the windward side of the cycle to the leeward one for TSR = 5 (Fig. 12) can be noticed. It seems that visualizations of flow structures for thin and thick profiles would help explain this phenomenon.

Fig. 12 clearly indicates a shift in the generation of mechanical energy from the windward half cycle to the leeward half cycle. Why? What is the reason for this behavior?

Is the sum correct?

Declines in tangential forces are observed in some phases of movement (windward side) and increases in others (leeward side). What are the effects within a single cycle? Are the turbine powers falling or rising? Or are they the same? What are the reasons for this? Why does a strongly cambered profile generate small tangential forces in the windward phase and large ones with the leeward?

The conclusions are correct, but drawn from other materials than those shown in the work. "• Although the NACA 0012 airfoil has the largest maximum lift coefficient of all symmetrical airfoils tested, it gives the worst results of the tangential blade load in the low tip speed ratio range. This is due to the worse airfoil characteristics in the detachment area compared is thicker airfoils. "

"• The analysis showed that symmetrical airfoils are much worse at low tip speed ratios. This is because of the worse characteristics of these airfoils in the stall regime. The introduction of one percent maximum camber greatly improves the aerodynamic performance of the rotor over the entire tip speed ratio range. " Will the cambering of a thin profile work similarly?

"• The effect of the relative airfoil thickness on the characteristics of aerodynamic blade load components is larger at low tip speed ratios, whereas, the maximum camber affects more these characteristics at higher tip speed ratios." Which chart has this conclusion been drawn from?

"• The study examined the impact of tip speed ratio on the velocity distribution in the aerodynamic wake of a rotor equipped with NACA0018 airfoils. Numerical analysis showed that as the tip speed ratio increases, there is a linear decrease in the average velocity $V_x$ (velocity component parallel to the wind direction) of these profiles. " This is a wake area, so isn't it obvious that receiving energy from the flow, as indicated by the increase in power factor in Fig. 11, must be reflected on the longitudinal velocity component in the wake.

Specific comments.

Fig. 4. No angle definition around the perimeter. It is in fig. 3.

Surrounding the 220 line unclear description. Line 220 - general information about Dx is given

It seems that Dx is incorrectly defined because it depends on TSR. Should it depend more on the speed? It is rather important how far the disturbance moves away at the flow speed to not affect the flow around the profile.

No 0.3. but 0.03

2.6.1 Unclear description

The definition of Re appears to be incorrect, no inflow speed.

240 - unclear sentence.

2.6.2 Unclear assumptions.

3.1. 3 blades?

Different chart scales (Fig. 10) do not allow checking the information given in the text. No zero axis.

What is the angle of attack corresponding to a 52 degree circumference angle?

The explanation for line 300 is good, but no hard arguments.

Mistake in description of Fig.12, is Normal should be Tangential

Fig. 11 and Fig. 13 - the form of the charts should be supplemented with charts containing lines for thickness and bending parameters. Existing charts are clear, but numerical values are difficult to read.

There is no graphic background - comparison of flow structures.

Perhaps the authors' assumption was to limit the information provided in the work to the description of changes in forces. They use software that can provide much more information about the flow and enable them to obtain answers to a number of questions that arose while reading the current version of the work.

The manuscript can be published in the form currently presented after removing minor errors.

---

## Author Comment (AC1) · 13 Mar 2020

Reply to Reviewer #1

Dear Reviewer,

We are very grateful for You to thoroughly review our work and for a number of valuable comments. We have read all your comments diligently and responded to them.

Specific comments:

**Reviewer:** Fig. 4. No angle definition around the perimeter. It is in fig. 3.

**Authors:** *Yes indeed, Currently, Figure 4 has been supplemented with the definition of azimuth.*

**Reviewer:** Surrounding the 220 line unclear description. Line 220 - general information about Dx is given

It seems that Dx is incorrectly defined because it depends on TSR. Should it depend more on the speed? It is rather important how far the disturbance moves away at the flow speed to not affect the flow around the profile.

**Authors:** *We corrected this sentence:*

The maximum values of the torque coefficient reach an almost constant values in each rotor revolution of about 1.3 after approximately 12 complete rotor revolutions.

**Reviewer:** No 0.3. but 0.03

**Authors:** *Yes, it is. Thank you for noticing this error.*

**Reviewer**: 2.6.1 Unclear description

**Authors:** *We have changed the description in such a way:*

RANS approach validation based on measured static data for NACA 0018

*we hope that it is now more understandable.*

**Reviewer:** The definition of Re appears to be incorrect, no inflow speed.

*Another part of the review also refers to the Reynolds number:*

The authors ignore the influence of the Reynolds number, but at constant wind speed, two TSR values mean two Re values at the same time. In the analyzed range of TSR values, this means a three-fold increase in Re between TSR = 2 and 6.

**Authors:** *In fact, it can be said that the influence of the Reynolds number was not considered in this work. For the Darrieus wind turbine, the definition of the Reynolds number is not an easy issue. The classic definition of the blade element Reynolds number is based on the relative velocity. This approach, however, requires extraction of the angle of attack from the velocity field around the blade for CFD and experimental research. Therefore, another Reynolds number is usually defined based on the tangential velocity of the blade:*

$$Re_1 = Re = \frac{\omega R\, c}{\nu}$$

*where: c is the chord length, R is the rotor radius and ω is an angular velocity of the rotor This definition is also used in our article (*==Equation 5==*).*

*Available sources sometimes use a definition of the Reynolds number (per unit) based on wind speed:*

$$Re_2 = Re_\infty = \frac{\rho_\infty V_0}{\mu_\infty}$$

*In general, there are works that address the problem of the influence of the Reynolds number on the performance of a Darrieus wind turbine. Most of this work uses the first mentioned definition of Reynolds number. The general conclusion from these investigations is that at constant rotor rotational velocity and variable wind speed (constant Reynolds number), an increase in the maximum aerodynamic efficiency of the rotor (maximum rotor power coefficient) is observed as the rotor rotational velocity increases. In this case, also the maximum aerodynamic efficiency shifts towards higher tip speed ratios as the RPM decreases. In our research, such a case was considered. Studies by other authors also show differences in the aerodynamic characteristics of the rotor if the wind speed is constant. Here, research shows that the rotor achieves higher maximum rotor power coefficients at lower wind speeds, however, at slightly lower tip speed ratios. In both cases, for constant RPM or for constant wind speed,*

*other factors such as the number of rotor blades have a significant impact. The effect of constant wind speed is much smaller than in the case of fixed RPM. However, one more aspect is important from the point of view of the task under consideration. The investigated wind turvine (McDONNELL 40-kW) was equipped with a system maintaining a constant RPM value on the shaft.*

*Below is a list of publications containing studies on the impact of the Reynolds number on rotor performance:*

B. F. Blackwell, R. E. Sheldahl, L. V. Feltz: Wind Tunnel Performance Data for the Darrieus Wind Turbine with NACA 0012 Blades. Research report. Sandia Laboratories SAND76-0130, 1976.
P. Bachant, M. Wosnik: Effects of Reynolds Number on the Energy Conversion and Near-Wake Dynamics of a High Solidity Vertical-Axis Cross-Flow Turbine. Energies, 9, 73, 2016.
R. Bogateanu, A. Dumitrache, H. Dumitrescu, C. I. Stoica: Reynolds number effects on the aerodynamic performance of small VAWTs. UPB Scientific Bulletin, Series D: Mechanical Engineering, 76, 1, pp. 25-36, 2014.
A. Rezaeiha, H. Montazeri, B. Blocken: Characterization of aerodynamic performance of vertical axis wind turbines: Impact of operational parameters. Energy Conversion and Management, 169, pp. 45-77, 2018.
S.-C. Roh, S.-H. Kang: Effects of a blade profile, the Reynolds number, and the solidity on the performance of a straight bladed vertical axis wind turbine. Journal of Mechanical Science and Technology, 27, 11, pp. 3299-3307, 2013.

*Let's compare three Reynolds numbers based on three different definitions. The third definition will be based on the relative velocity calculated for a simple theoretical model. In this model, the relative velocity $V_{rel}$ is the geometric sum of the tangential velocity vector of the blade (with a minus sign) and the vector of wind speed, $V_0$:*

$$V_{rel} = \sqrt{(\omega R + V_0 \cos\theta)^2 + (V_0 \sin\theta)^2}$$

*The local angle of attack is expressed by the formula:*

$$\alpha = \tan^{-1}\left(\frac{V_0 \sin\theta}{\omega R + V_0 \cos\theta}\right)$$

*Then the Reynolds number is expressed by the definition:*

$$Re_3 = \frac{\rho_0 V_{rel}}{\mu_0}$$

*Let's use geometric data and operating parameters of our rotor:*

R=8.48 m
$\omega$=8.18 rad/s
$V_0$=13.4 m/s
$\mu_0$=1.79E-05 kg/ms
$\rho_0$=1.225 kg/m$^3$
c=0.61 m

*We obtain the following results for these parameters:*

| Parameter | Value |
|---|---|
| Tip speed ratio, TSR | 5.2 |
| Reynolds number, $Re_1$ | 289 6728 |
| Reynolds number, $Re_2$ | 559 390 |

[Figure]

[Figure]

**Reviewer:** 240 - unclear sentence.

**Authors:** *We corrected this sentence, it is now:*

This Reynolds number was used to determine the undisturbed flow velocity $V_0$ for steady-state simulations. The undisturbed flow velocity in RANS simulations is equal to the tangential velocity of the rotating rotor blade, $V_0 = \omega R$. In these investigations, it was assumed that the angular velocity of the rotor was constant for all tip speed ratios and equal to 8.18 rad / s. Knowing that the rotor radius is 8.48 m, the tangential velocity of the rotor blade is 69.4 m / s for the Reynolds number to be equal to 2.9 million.

**Reviewer:** 2.6.2 Unclear assumptions.

**Authors:** *We have improved this chapter*

*Initial:*

All numerical results presented in this paper were computed using ANSYS Fluent solver and the unsteady RANS (URANS) method with the SST k-ω turbulence model. For the comparison, the aerodynamic blade loads were also computed using a CFD code FLOWer from German Aerospace Center that was continuously extended for wind energy applications at IAG (Weihing et al., 2019) and a vortex model from Technical University of Denmark (DTU). The vortex method

calculates the induction at the blade positions from the influence of the continoulsy developing wake vorticity using the Biot-Savart law. The strength of the instantaneous bound circulation was calculated from the local inflow and airfoil data using the Kutta-Joukowski theory and a new vortex was released in every time step at the trailing edge to ensure that the total circulation remains constant in time. The airfoil data were calculated using XFoil and presented in Fig. 8. However, the effect of decambering from the blades moving in a curved was estimated as approximately 1 degree using thin airfoil theory and effectively shifting the lift curve 1 degree to the left. No dynamic stall model was used here. In these investigations, the rotor operating at the tip speed ratio of 5 and equipped with NACA 0018 airfoils was analysed. The normal and tangential blade loads are shown in Fig. 9. The two employed CFD codes and the vortex model deliver similar results of aerodynamic blade loads and taken as verification that the equations are solved correctly.

*To:*

All numerical results presented in this paper were computed using ANSYS Fluent solver and the unsteady RANS (URANS) method with the SST k-ω turbulence model. For the comparison, the aerodynamic blade loads were also computed using the CFD code FLOWer from German Aerospace Center that was continuously extended for wind energy applications at IAG (Weihing et al., 2019) and a vortex model from Technical University of Denmark (DTU).

The FLOWer code is a compressible code based on a fully structured mesh approach. Therefore, a different mesh was adopted in the calculations, employing an overset (Chimera) grid technique. This enables the user to built high quality meshes of each component independently. Further information about the mesh strategy and its topology is given in Bangga et al. (2017b). The mesh of the rotor blade is resolved by 320 grid cells in the circumferential direction and by 32 cells across the boundary layer with a growth rate of 1.17, while maintaining the non-dimensional wall distance of y+ < 1.0. This mesh component is combined with fully structured Cartesian background meshes having a grid cell size of 0.23 m ($\Delta \approx 0.026R$) near the rotor, based on a grid study for wake flow statistics in Bangga et al. (2017b). The computations were carried out assuming a fully turbulent boundary layer employing the Menter SST k-w model, consistent with the simulations performed using ANSYS Fluent.

The vortex method calculates the induction at the blade positions from the influence of the continuously developing wake vorticity using the Biot-Savart law. The strength of the instantaneous bound circulation was calculated from the local inflow and airfoil data using the Kutta-Joukowski theory. A new vortex released in every time step at the trailing edge ensures that the total circulation

remains constant in time. The airfoil data were calculated using XFoil and presented in Fig. 8. However, the effect of decambering from the blades moving in a curved was estimated as approximately 1 degree using thin airfoil theory and effectively shifting the lift curve 1 degree to the left. No dynamic stall model was used here. In these investigations, the rotor operating at the tip speed ratio of 5 and equipped with NACA 0018 airfoils was analysed. The normal and tangential blade loads are shown in Fig. 9. The two employed CFD codes and the vortex model deliver similar results of aerodynamic blade loads and taken as verification that the equations are solved correctly. However, some discrepancies are still observed in the predicted forces. FLOWer and Fluent deliver very similar predictions, while the forces obtained using the vortex method are slightly smaller. It shall be noted that the vortex method relies on the static polar data as its input. The results indicate the effects of Reynolds number variation over the azimuth and the rotor rotation on the blade boundary layer development. Despite that, the studies also demonstrate that all employed approaches are consistent for predicting the general characteristics of the rotor performance.

Bangga, G., Lutz, T., Dessoky, A. and Krämer, E.: Unsteady Navier-Stokes studies on loads, wake, and dynamic stall characteristics of a two-bladed vertical axis wind turbine, Journal of Renewable and Sustainable Energy, 9, 053303, https://doi.org/10.1063/1.5003772, 2017b.

**Reviewer:** 3.1. 3 blades?

**Authors:** *We have changed the title of this section. We hope that now it is clearer that we mean the distribution of loads on one rotor blade. The current name of chapter 3.1 is:*

3.1 Aerodynamic loads on a rotor blade for symmetrical 4-digit NACA airfoils

**Reviewer:** Different chart scales (Fig. 10) do not allow checking the information given in the text.
No zero axis.

**Authors:** *Yes, that's a very good point. Currently, in Figure 10, the scales are the same for two load components. In addition, we have added the y-axis = 0 for all four cases.*

**Reviewer:** What is the angle of attack corresponding to a 52 degree circumference angle?

**Authors:** *Following the considerations of the geometric angle of attack presented above and below, we introduced the following changes in two parts of the article:*

1. *We corrected the part of the paper:*

The geometric angle of attack for a tip speed ratio of 2 is 16.7 deg. The geometric angle of attack model assumes that the flow velocity at the rotor is equal to wind speed. In fact, the velocity at the rotor is lower compared to the wind speed. Therefore, the actual angle of attack will be lower than this geometric angle of attack. Nevertheless, a 52 degree azimuth is located in the upwind part of the rotor. The wind seed is not so strongly slowed down in this part of the rotor compared to the downwind part of the rotor (Paraschivoiu, 2002). Therefore, the difference between the real and geometric angles of attack should not be large. Comparing this conclusion with the lift force coefficient for the NACA 0012 airfoil shown in this response letter, it can be stated that flow detachment and loss of lift occurs at this azimuthal location.

*The following source was used to answer this question:*

Paraschivoiu, I.: Wind turbine design: with emphasis on Darrieus concept, Presses inter Polytechnique, 2002.

2. *The authors made the following changes in the manuscript:*

Analyzing steady-state experimental data of lift coefficients of NACA 0012, NACA 0015 and NACA 0018 obtained by Sheldahl and Klimas (1981) for Reynolds number of $1 \cdot 10^6$ it can be stated that the airfoil with a relative thickness of 12% achieves the highest value of the maximum lifting force coefficient, however, the critical angle of attack is smaller compared to the NACA 0018 airfoil. If the blade pitch angle is zero, the geometric angle of attack is measured between the relative velocity and the wind velocity. At the tip speed ratio of 2 the azimuthal angle of 52 deg. corresponds to the local geometric angle of attack of 16.7 deg. The effective angle of attack is of course lower than the geometric angle of attack because the local flow velocity at the rotor is lower than the wind speed. However, this difference is not very large, especially in the upwind part of the rotor (Paraschivoiu, 2002). Therefore, for the NACA 0012 airfoil and at the azimuth of 52 deg. flow detachment and consequent loss of lift may be expected. In addition, in the case of the NACA 0012 airfoil, the decrease in lift above the critical angle is much more rapid, indicating a leading edge stall, than in the case of NACA 0015 and NACA 0018 airfoils, for which the decrease in lift is much milder and where the separation starts at the trailing edge and gradually increasing with the angle of attack.

**Reviewer:** Mistake in description of Fig.12, is Normal should be Tangential

**Authors:** *We are very grateful for this comment. Actually, there was a mistake. It has already been corrected.*

**Reviewer:** Fig. 11 and Fig. 13 - the form of the charts should be supplemented with charts containing lines for thickness and bending parameters. Existing charts are clear, but numerical values are difficult to read.

**Authors:** *We made such drawings, but it turned out that in this case contour maps will provide better readability. Please see the sample $C_P$ (TSR) drawing for symmetrical profiles of different thicknesses.*

[Figure]

*And for comparison, the contour map:*

[Figure]

**Reviewer:** There is no graphic background - comparison of flow structures.

*also*

Interesting work. Focused on the description of the impact of changes in thickness and camber of the H-turbine blade profile on its aerodynamic properties. Changes to these characteristics are discussed without analyzing changes in the flow structure. Perhaps a separate work is being devoted by the Authors to the analysis of changes in flow structures.

*also*

To explain the reasons for the change in turbine performance characteristics, it may be necessary to pay attention to the following effects. For TSR = 2, the angle of attack change is + - 30 degrees, which means fow separation. For TSR = 5, the max angle of attack change is + - 11.6 degrees, which means work without flow separation. For TSR = 2, with the blade rotation angle of 46 degrees, the profile approach angle of attack 15 degrees corresponding to the flow separation. For TSR = 3, at a blade rotation angle of 66 degrees, the angle of attack of the profile exceeds 15 degrees corresponding to the flow separation. For TSR = 4 and larger TSRs, the profile angle of attack never exceeds the critical angle of attack. The

influence of flow inertia effects is not strong. Increasing TSR means only slightly increasing the rate of the angle of attack changes.

**Authors:** *Indeed, that is a good point. In our work, we compared many cases focusing on the aspect of forces and velocity distributions. Each of the examined cases has a different distribution of instantaneous flow parameters. This is of course a very interesting issue, but we think that a very detailed analysis of these parameters will cause that our paper can grow significantly and thus it may become less clear. We believe that it is better to devote a separate paper to this issue.*

*A geometric angle of attack is created on the basis of a velocity triangle made up of vectors of wind speed, tangential velocity of the blade and relative speed. In this simple model, the angle of attack depends only on two parameters: azimuth and tip speed ratio. Chart of the maximum angle of attack as a function of tip speed ratio is as follows:*

[Figure]

*Using the aerodynamic characteristics for the NACA 0018 airfoil (Figure 8 in the Manuscript) it can be seen that for TSR ≥ 4 flow separation on the blades of a rotating rotor should not occur, as the reviewer rightly pointed out.*

In chapter 3.1 we showed the physical cause of changes in tangential force for TSR = 2. We added the following figures:

[Figure]

**Figure 11: Static pressure (in Pascal) around the NACA 0012 airfoil at the rotor tip speed ratio of 2.**

[Figure]

**Figure 12: Static pressure (in Pascal) around the NACA 0018 airfoil at the rotor tip speed ratio of 2.**

**Reviewer:** It is seen lack of basic characteristics for different profile thicknesses, although the authors provide some information. "In addition, in the case of the NACA 0012 airfoil, the decrease in lift above the critical angle is much more rapid, indicating a leading edge stall, than in the case of NACA 0015 and NACA 0018 airfoils, for which the decrease in lift is much milder and where the separation starts at the trailing edge and 300 gradually increasing with the angle of attack. "

**Authors:** *In our article we have only shown the characteristics of lift and drag for one airfoil NACA 0018. We did it for two reasons:*

- *The original McDonnell 40-kW Giromill was equipped with NACA 0018 airfoil blades.*
- *Due to better transparency of the entire article, whose main topic is transient blade characteristics.*

*Using the experimental data by Sheldahl and Klimas (1980), available in the Internet, the following figure shows the missing airfoil characteristics of three airfoils: NACA 0012 and NACA 0018 (this is the only data we have). These characteristics are given for two Reynolds numbers, 2 million and 5 million. These experimental data show a rapid drop in lift for a thin airfoil.*

E. Sheldahl, P. C. Klimas, P. C.: Aerodynamic Characteristic of Seven Symmetrical Airfoil Sections Through 180-degree Angle of Attack for Use in Aerodynamic Analysis of VAWT. Technical Report SAND80-2114. Albuquerque, New Mexico, USA: Sandia National Laboratories, 1980.

[Figure]

Changes in the manuscript:

*In chapter 2.6.1 before Figure 8 we added the following supplement to symmetrical profiles:*

Experimental and CFD investigations show that the course of the lift coefficient is almost identical for symmetrical airfoils, NACA 0012, NACA 0015 and NACA 0018, in the range of the angle of attack from 0 to the critical angle of attack, $\alpha_{crit}$. Above this critical angle of attack, the differences in lift characteristics are visible. Analyzing the aerodynamic derivative $dC_L/d\alpha$ in the range of the angle of attack from $\alpha_{crit}$ to 18 degrees (the maximum angle of attack analyzed using the RANS approach) it was found that for the NACA 0012 airfoil it is -0.11, for the NACA 0015 airfoil it is equal to -0.03 while for the NACA 0018 airfoil it is -0.005 (Sheldahl and Klimas 1981). In the case of CFD analysis, these derivatives are respectively: -0.17, -0.033 and -0.0051. This analysis shows that for the NACA 0018 and NACA 0015 airfoils, the decrease in lift force is milder compared to the NACA 0012 airfoil.

*In the Section 3.1 we added the following sentence (underlined in yellow):*

Analyzing steady-state experimental data of lift coefficients of NACA 0012, NACA 0015 and NACA 0018 obtained by Sheldahl and Klimas (1981) for Reynolds number of $1\cdot10^6$ it can be stated that the airfoil with a relative thickness of 12% achieves the highest value of the maximum lifting force coefficient, however, the critical angle of attack is smaller compared to the NACA 0018 airfoil (this aspect was also discussed in Section 2.6.1).

**Reviewer:** The dramatic change in the performance characteristics for TSR = 2 is surprising. When the camber changes, the transfer of received energy from the windward side of the cycle to the leeward one for TSR = 5 (Fig. 12) can be noticed. It seems that visualizations of flow structures for thin and thick profiles would help explain this phenomenon.

Fig. 12 clearly indicates a shift in the generation of mechanical energy from the windward half cycle to the leeward half cycle. Why? What is the reason for this behavior?

**Authors:** *Explanation of Fig. 12 (presently Fig. 14)*

*This drastic decrease in tangential force for the NACA 0018 airfoil may also be associated with the so-called camber effect. Blades on the Darrieus wind turbine are subjected to the curved flow filed that was caused by the revolution of wind*

*turbine rotor. Therefore, the flow around the rotor with symmetrical airfoil looks like this:*

[Figure]

*In that case the lift coefficient $C_L(\alpha)$ will look as:*

[Figure]

*And the lift will be reduced on the upstream part od the rotor and increased at the downwind part compared to a non-cambered airfoil. This is consistent with Fig. 12 (now Fig. 14).*

*The following explanation has been introduced into the article (Section 3.2):*

The decrease in the tangential blade load for the NACA 0018 airfoil (Fig. 14) can also be caused by so-called virtual blade camber. It is caused by a curved flow field caused by the revolution of the wind turbine rotor. Airfoil characteristics are usually measured in a wind tunnel with straight flow field. The curved flow field causes that the symmetrical airfoil of a vertical axis wind turbine works as a cambered airfoil. This causes a change in the lift coefficient characteristics of the airfoil. For a given angle of attack, the value of the lift coefficient of the

symmetrical airfoil with the "virtual camber" is lower compared to the symmetrical airfoil in straight flow field.

Akimoto, H., Hara, Y., Kawamura, T., Nakamura, T. and Lee, Y-S.: A conformal mapping technique to correlate the rotating flow around a wing section of vertical axis wind turbine and an equivalent linear flow around a static wing, Environmental Research Letters, 8, https://doi.org/ 10.1088/1748-9326/8/4/044040, 2013.

---

## Referee Comment (RC2) · Anonymous Referee #2 · 15 Apr 2020

Performance analysis of a Darrieus-type wind turbine for a series of 4-digit NACA airfoils

The authors present results of a study of aerodynamic performance analysis of an H-VAWT by varying airfoil thickness and camber for NACA airfoils. In general, the results show trends as expected with airfoil thickness, TSR, etc. However, it's not clear the new contribution given existing published work that has not been referenced. The authors should provide some clarification on the novelty points of their paper and update their literature review.

Comments: 1. The authors based their study on the McDonnell 40kW machines. Was any comparison to experimental data from this machine performed? Why was this machine selected over others VAWT testbeds having rich data sets for comparison?

2. In conclusion, the authors note that CFD has better agreement with experimental polars for Cl values and XFOIL has better agreement for Cd values. Can some explanation for this be provided? 3. The research in this paper is very dependent on the aerodynamic polar data. But, there is no exhaustive polar analysis of all airfoils (including symmetric and cambered airfoils) used in the analysis to understand the general trends and stall performance of all airfoils. However, such a detailed analysis has already been done in other published works not referenced in this manuscript (see next point). 4. The following two works appear to present all or most of the findings in the present manuscript, in particular the first paper. These papers present similar but comprehensive numerical and experimental studies for VAWT aerodynamic performance: 4.1. "Investigation on aerodynamic performance of vertical axis wind turbine with different series airfoil shapes" Renewable Energy 2018. Link to paper follows. * https://www.sciencedirect.com/science/article/pii/S0960148118302398 4.2. "Study on stall behavior of a straight-bladed vertical axis wind turbine with numerical and experimental investigations" Journal of Wind Energy and Industrial Aerodynamics 2017. Link to paper follows: * https://www.sciencedirect.com/science/article/pii/S016761051630174X 5. Following point #4 above, the authors should clarify the novelty points of their work versus these works and other relevant similar works.